# Predicting long-term functional anti-VEGF treatment outcomes in neovascular AMD in a real-world setting

Isabel B. Pfister[1], Christin Schild[1], Justus G. Garweg[1,2]*

1 Swiss Eye Institute and Clinic for Vitreoretinal Diseases, Berner Augenklinik, Bern, Switzerland,
2 Department of Ophthalmology, Inselspital, Bern University Hospital, University of Bern, Bern, Switzerland

* justus.garweg@augenklinik-bern.ch

**Data Availability Statement:** All data used for the analyses, as well as the R code for the regression analyses, can be found here: https://osf.io/7fgmk/ The DOI is: 10.17605/OSF.IO/7FGMK. All study findings can be replicated using this data.

## Abstract

### Purpose

To test to what degree retinal fluid (RF) after the loading phase and at the end of year 1 predicts long-term functional outcomes in neovascular macular degeneration (nAMD), as do macular (MA) atrophy, treatment density and treatment interval extension.

### Methods

In this retrospective single-center cohort study, a consecutive series of eyes with treatment-naïve nAMD followed under a treat-and-extend (T&E) protocol followed over ≥2 years. Best-corrected visual acuity (BCVA), presence of retinal fluid (RF) and macular atrophy (MA) were registered along with central retinal thickness (CRT) and treatment density over time. The relationship between these variables was tested by regression analysis.

### Results

A total of 433 eyes were followed for 4.9 ± 2.2 years. A series of univariate analyses were run to select the covariates for the final multivariate regression model. CRT after loading, time to dryness, intraretinal fluid and MA after one year were found to predict visual function over 2 to 5 years. A final regression model was adjusted for visual acuity (VA) at baseline and showed that CRT after loading was predictive only in the short term (2 years) and that MA had the greatest predictive value for VA after 2 to 5 years. Intraretinal fluid (IRF) significantly predicted VA only after 4 years. The final regression model explained 21 to 32% of the variation in VA.

### Conclusions

In this large retrospective cohort, the presence of MA after one year was the strongest predictor of VA after 2 to 5 years, explaining a vision loss of 13 to 20 letters. The presence of IRF and SRF at any point of time had a comparably weak predictive potential for the outcomes over 5 and more years.

**Funding:** The author(s) received no specific funding for this work.

**Competing interests:** None of the authors received direct or indirect support for this investigator-initiated study or have any conflicting interests with the data that are presented in this report.

## Introduction

Several factors have been reported to contribute to the functional and anatomic prognosis of neovascular AMD (nAMD) patients receiving anti-VEGF treatment. Low baseline visual acuity (VA), preexisting structural damage to the macula, submacular hemorrhage and lesion size may be beyond the non-influenceable outcome predictors [1]. VA after the first year is closely linked to early diagnosis and treatment initiation [1–7]. Due to a ceiling effect, patients with a baseline VA above 20/40 are expected to have a lower visual gain but, on the other hand, will have the best final visual outcomes over time [8, 9]. Due to a ceiling effect, patients with excellent baseline VA may not gain but may even lose single letters, but attain an excellent functional outcome, whereas patients with very low visual acuity (below 20/100) may gain more than 15 letters; nevertheless, they do not reach reading or driving vision over 2 years [10] or up to 5 years [11]. On the other hand, the baseline VA may be less predictive of visual potential than the baseline VA after the loading phase [9, 12, 13].

Persistent disease activity, as evidenced by the presence of IRF and subretinal fluid (SRF), may result from insufficiently intensive early treatment [14] or incomplete treatment response. Persistent fluid after the loading phase is linked to a less favorable functional outcome even in eyes without significant structural damage [15]. Up to 66% of patients, depending on the anti-VEGF drug used, exhibit persistent disease activity after the loading phase in a real-world setting [16]. If complete control of disease activity is not achieved after the loading phase, a visual decline is expected within 12 to 48 months (PRN study arms of the Harbor [17], Catt [18–20], Montblanc [21]), whereas consequent suppression of VEGF activity and thereby prevention of lesion growth will result in visual stability for up to 5 years (fixed treatment arms in the Anchor [22], Marina [23], Harbor [24], CATT [20] and View studies [25]). However, disease stability (absence of intraretinal fluid), which is achieved in two-thirds of eyes after one year, will be maintained at only 53% over 2 years [26]. Independent of treatment, macular atrophy can develop or progress in virtually any patient at some point [22].

Given the aforementioned evidence, we wished to model the long-term functional outcomes of anti-VEGF therapy after correcting for baseline VA in a treatment-agnostic approach based on VA and morphological parameters in eyes receiving three loading injections and followed for at least two years.

## Patients and methods

In this single-center retrospective cohort study, we included patients with treatment-naïve nAMD who began receiving intravitreal treatment (IVT) with ranibizumab (Ran) or aflibercept (Afl) between October 2012 and October 2021 under a T&E protocol that included three initial loading injections at monthly intervals and who were followed for at least two years. According to this protocol, intervals were adopted with 2-week in- or decrements based on OCT findings: Interval extension was allowed in the absence of intraretinal and subretinal fluid, or if subretinal fluid and PED had remained unchanged over 3 consecutive visits; shortening of the interval was requested in case of new or increasing fluid in any compartment or new retinal hemorrhage indicating disease activity. Data were collected between October 2023 and January 2024. Eyes with preexisting structural damage preventing a functional recovery and major submacular hemorrhages were excluded from the analysis. Furthermore, patients needed to have at least four clinical visits as well as at least five IVTs in the first year. Best-corrected distance visual acuity (BCVA) was transformed into early treatment of diabetic retinopathy score (ETDRS) values, where a Snellen decimal BCVA of 1.0 was defined as 85 ETDRS letters. Furthermore, OCT data including CRT in all eyes and the presence of IRF and/or SRF were retrospectively collected from the patient records at the following time points: at baseline,

at three and 12 months and annually thereafter along with the treatment demand and intervals over time. MA at the end of year 1 was not quantified but was categorized as absent or present if incomplete (iRORA) or complete RPE and outer retinal atrophy (cRORA), incomplete (iORA) or complete outer retinal atrophy (cORA) were identified in the foveal center. These criteria were based on specific OCT findings: (1) a foveal region of hypertransmission of at least 250 μm in diameter, (2) a zone of attenuation or disruption of the RPE of at least 250 μm in diameter, (3) evidence of overlying photoreceptor degeneration, and (4) absence of scrolled RPE or other signs of an RPE tear [27]. Furthermore, the time to dryness was deducted from the clinical records, and the treatment intervals were recorded. From this, we calculated how many eyes could extend treatment to ≥10 weeks within the first two years of therapy. In contrast to the majority of clinical trials, treatment was considered extended if at least three consecutive intervals were ≥10 weeks.

The study was approved by the Ethics Committee of the Canton of Bern (KEK no. 2022–01850) and is fully compliant with the tenets of the Declaration of Helsinki in its latest version. Each participant provided a general written consent for the use of their coded clinical data.

All variables are measured at the between subject level. No outliers were eliminated for any of the analyses. Multiple regression analyses were applied to predict short- (2 years) and long-term (3–5 years) therapy responses. The data are presented as the mean ± standard deviation (SD), median and 25% and 75% interquartile ranges (IQRs), unless otherwise stated. The level of significance was set at $p < 0.05$. All the statistical analyses were performed using R (software version 4.3.2). We chose VA at baseline, change in VA from baseline to the end of the loading and to 1 year of therapy as possible control variables for the regression analyses. CRT after loading, change in CRT from baseline to the end of loading, and to 1 year, SRF after loading or after 1 year, IRF after loading or after 1 year, and IRF as well as SRF after loading or after 1 year, time to dryness, interval extension to > 10 weeks in the first year or in the second year, macular atrophy at baseline, after loading and after the first year and number of injections after the first year were considered as possible independent variables. Out of these parameters, for our multivariate analyses we chose the ones which maximally explained the variance in all cases.

All data used for the analyses, as well as the R code for the regression analyses, can be found here: https://osf.io/7fgmk/

The DOI is: 10.17605/OSF.IO/7FGMK. All study findings can be replicated using this data.

## Results

A total of 433 eyes of 404 consecutive patients admitted to our clinic for the treatment of their neovascular AMD between 2012 and 2021 met the inclusion criteria. The demographic information of the participants is presented in **Table 1**. Of the 433 eyes, 228 (52.7%) were pseudophakic before their first anti-VEGF Treatment, 108 (24.9%) underwent a cataract surgery during follow up. Cataract surgery during follow up had no influence on the development of MA after 1 year ($r = 0.052$, $p = 0.30$), and did not impact visual acuity at any time point in these 108 eyes compared to the rest of the sample. A total of 39.5% of the sample was lost to follow-up. Of these, 15.3% moved away, 11% died, and 3% reached stability and decided to stop treatment. Another 3% paused treatment because of systemic illness, whereas 2% stopped treatment because of the loss of a visual potential. One percent of the patients wished to discontinue treatment, and the remaining 4.2% stopped treatment for unknown reasons. The findings for VA are displayed in **Figs 1–3**: Fig 1 shows the changes in VA from baseline, stratified in three groups based on baseline visual acuity (BCVA < 0.2, n = 82; BCVA 0.25–0.5, n = 217; BCVA > 0.5, n = 133). The first two groups experienced an increase in VA after the

**Table 1. Epidemiological data at baseline.**

| | Sample (n = 433 eyes, 404 patients) |
|---|---|
| | *Epidemiological data* |
| Age (years: mean ± SD; min-max) | 79.1 ± 8.3; 50.0–100.5 |
| Gender (% females) | 63.7 |
| Pseudophakia (%) | 52.7 |
| Total follow-up time (years: mean ± SD; min-max) | 4.9 ± 2.2; 2.0–10.7 |
| Still under therapy at the end of study (%) | 60.5 |

[#] Chi-square test and Mann-Whitney U test were applied to test for intergroup differences.

loading phase and remained stable thereafter. The group with the best baseline VA experienced a less pronounced VA increase after baseline, but remained stable over the entire follow up period. Group comparisons between the three groups (Kruskal-Wallis H test) showed that the change in VA was different between the three groups from baseline to years one, two and three (all $p < 0.001$). For the change from baseline to years four and five, a difference was found between group <0.2 and the other two groups ($p < 0.05$). Fig 2 displays eyes with IRF after one year with those who did not, regarding the change in VA over time. Fig 3 compares the change in VA over time in eyes without and with MA after one year. Eyes with MA

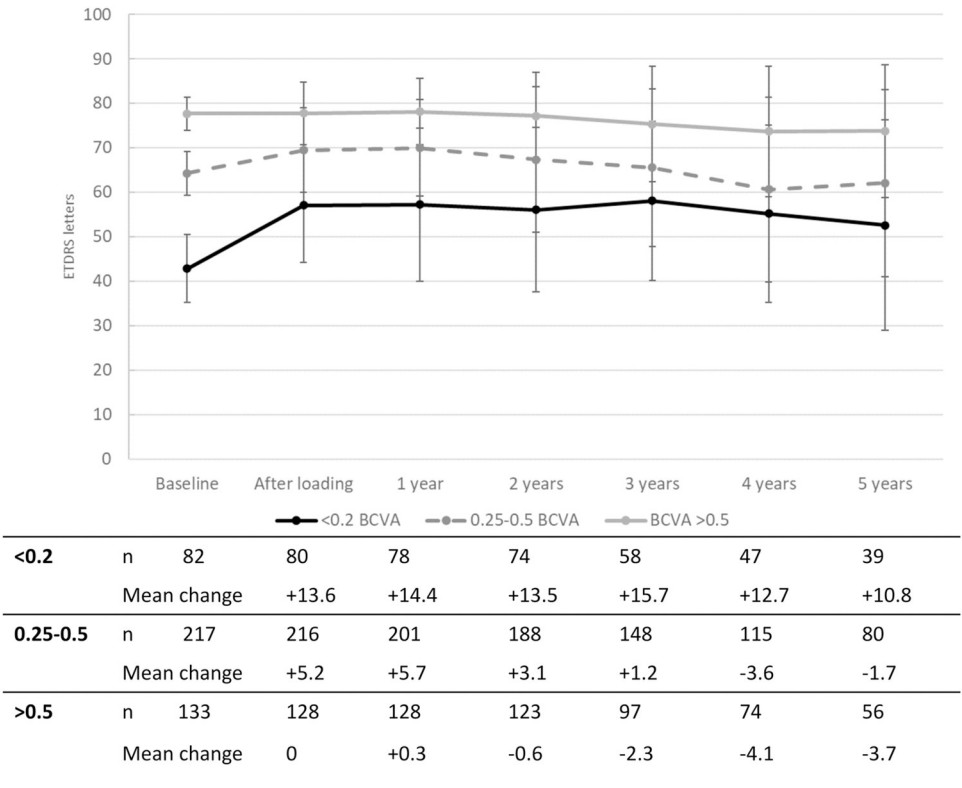

| | | Baseline | After loading | 1 year | 2 years | 3 years | 4 years | 5 years |
|---|---|---|---|---|---|---|---|---|
| **<0.2** | n | 82 | 80 | 78 | 74 | 58 | 47 | 39 |
| | Mean change | | +13.6 | +14.4 | +13.5 | +15.7 | +12.7 | +10.8 |
| **0.25-0.5** | n | 217 | 216 | 201 | 188 | 148 | 115 | 80 |
| | Mean change | | +5.2 | +5.7 | +3.1 | +1.2 | -3.6 | -1.7 |
| **>0.5** | n | 133 | 128 | 128 | 123 | 97 | 74 | 56 |
| | Mean change | | 0 | +0.3 | -0.6 | -2.3 | -4.1 | -3.7 |

**Fig 1. Change in visual acuity (VA, in ETDRS letters) compared to baseline, stratified by baseline VA (Decimal BCVA <0.2; 0.25–0.5; >0.5).**

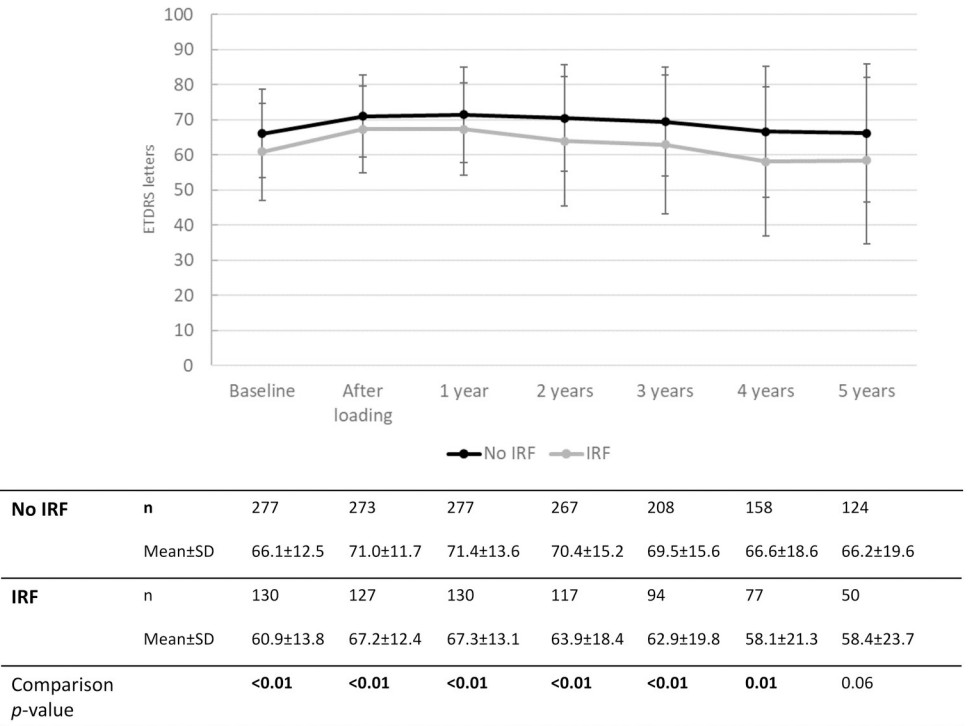

| No IRF | n | 277 | 273 | 277 | 267 | 208 | 158 | 124 |
|---|---|---|---|---|---|---|---|---|
| | Mean±SD | 66.1±12.5 | 71.0±11.7 | 71.4±13.6 | 70.4±15.2 | 69.5±15.6 | 66.6±18.6 | 66.2±19.6 |
| IRF | n | 130 | 127 | 130 | 117 | 94 | 77 | 50 |
| | Mean±SD | 60.9±13.8 | 67.2±12.4 | 67.3±13.1 | 63.9±18.4 | 62.9±19.8 | 58.1±21.3 | 58.4±23.7 |
| Comparison p-value | | <0.01 | <0.01 | <0.01 | <0.01 | <0.01 | 0.01 | 0.06 |

**Fig 2. Visual acuity over time stratified by the presence of intraretinal fluid (IRF) after the 1st year.**

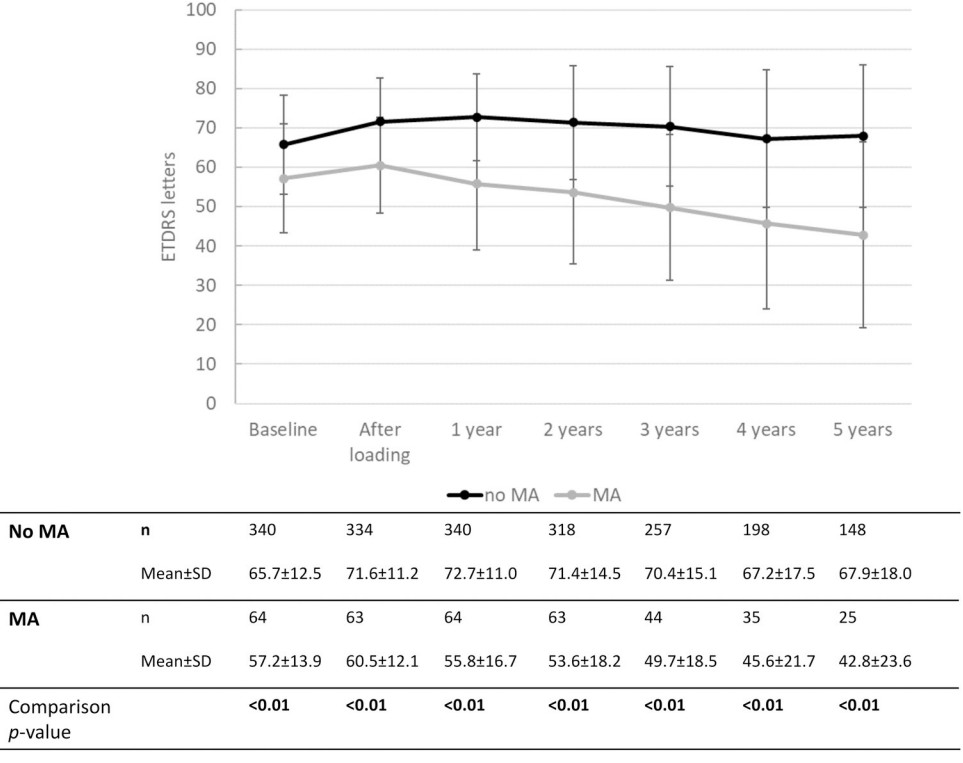

| No MA | n | 340 | 334 | 340 | 318 | 257 | 198 | 148 |
|---|---|---|---|---|---|---|---|---|
| | Mean±SD | 65.7±12.5 | 71.6±11.2 | 72.7±11.0 | 71.4±14.5 | 70.4±15.1 | 67.2±17.5 | 67.9±18.0 |
| MA | n | 64 | 63 | 64 | 63 | 44 | 35 | 25 |
| | Mean±SD | 57.2±13.9 | 60.5±12.1 | 55.8±16.7 | 53.6±18.2 | 49.7±18.5 | 45.6±21.7 | 42.8±23.6 |
| Comparison p-value | | <0.01 | <0.01 | <0.01 | <0.01 | <0.01 | <0.01 | <0.01 |

**Fig 3. Visual acuity over time stratified by the presence of macular atrophy after the 1st year (MA).**

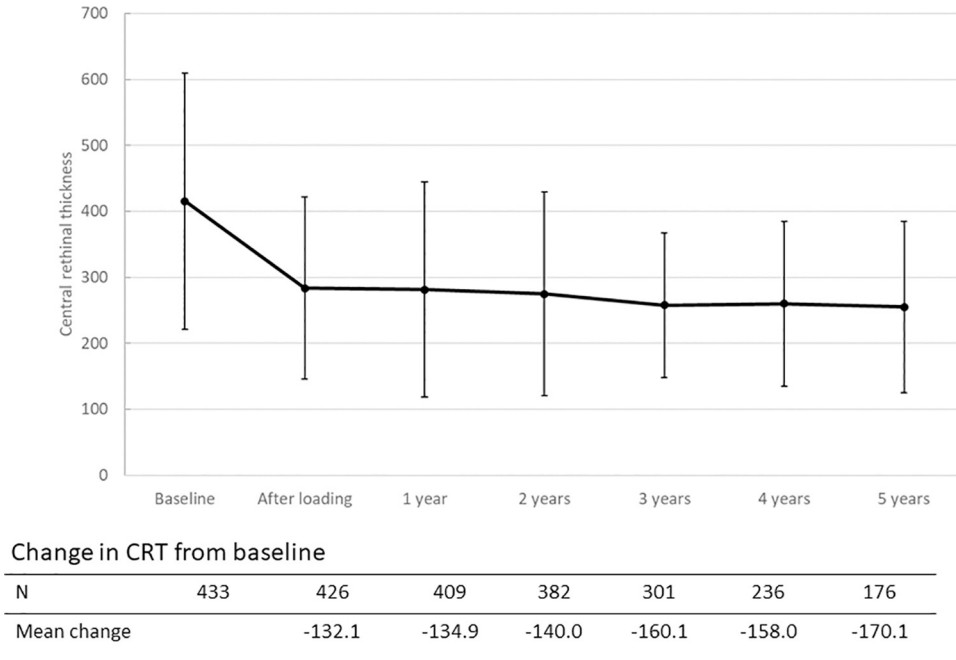

**Fig 4. Changes in central retinal thickness (CRT) during the treatment period up to 5 years of follow up.**

experienced a decrease in VA over time, whereas eyes without MA remained stable over 5 years after the loading. The course of CRT and its change over time compared to baseline are displayed in Fig 4.

### Retinal fluid

The portion of eyes with retinal fluid at the different time points is displayed in Fig 5. After the loading phase, almost half of the sample had a dry retina (48.5%), 16.4% of the eyes showed IRF, 21.2% SRF, and 12.2% had both IRF and SRF.

### Treatment interval extension

Within the total sample of 433 eyes, a mean treatment interval extension to 11.1 ± 3.5 weeks (median: 11.0, IQR: 9–13) was registered. The mean time to reach ≥10 weeks was 15.5 ± 13.9 months (median: 10.1, IQR: 6.9–19.2; range: 8–83.9 months; S1 Fig). The treatment interval was extended to ≥10 weeks over three consecutive visits within the first two years of therapy in half of the sample (50.8%).

### Multiple regression analysis

We first performed a series of univariate regression analyses to identify covariates with predictive potential for short- (2 years) and long-term (3–5 years) VA. We examined CRT levels after loading and compared them to the change in CRT from baseline to after loading or the change from baseline to the first year. We also examined the presence of IRF, SRF, and RF after the loading phase and after the first year, as well as the time to dryness and interval extension to ≥10 weeks within the 1st and 2nd years. Finally, we examined MA at baseline, after the loading phase and after one year as well as the number of injections in the first year as covariates for the prediction of VA (Table 2). Covariates with a significant predictive value over at least 2

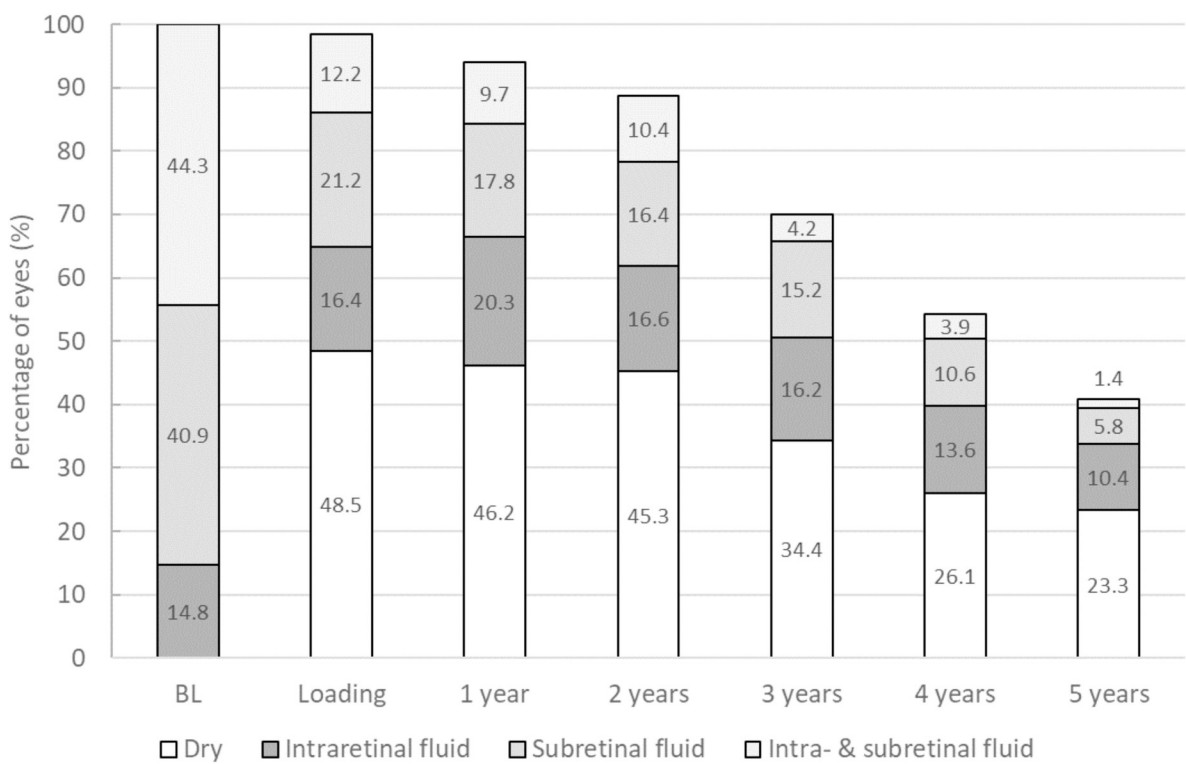

**Fig 5. Presence of retinal fluid during the treatment period up to 5 years of follow up.**

timepoints were integrated into a multivariate model in the second step. In cases with several options for the same variable (CRT, IRF and SRF as well as MA), the timepoint with the most explanatory potential ($R^2$) was chosen. Here, we found that CRT after loading, IRF after the 1st year, time to dryness and MA after 1 year had the strongest predictive potential for VA after 2 to 5 years. MA at baseline was not predictive for VA at any time point, explaining maximally 2% of the variation. MA after the loading was significant only for VA at 3 years, explaining 8% of the variation in VA, while MA at year 1 explained 18% of the variation. Interval extension to ≥10 weeks was predictive of short-term, but not long-term VA (up to 3 years). We therefore decided not to include this covariate in our model. The effect sizes of the variables which were included into the final model are displayed in S1 Table.

We also examined which was the best control variable, VA at baseline, compared to the change in VA from baseline to after loading or after one year. Here, we found that VA at baseline was a stronger predictor of long-term VA than change in VA.Therefore, we included VA at baseline as a control variable in our model.

In the second step, we constructed a model containing all covariates which had been identified in univariate analyses to have predictive potential for VA after control for VA at baseline (Table 3). The confidence intervals for the covariates in the models are reported in S2 Table. In this regression model, we found that interval extension to ≥10 weeks within the first 2 years of treatment was predictive for VA at years 2–4, but not thereafter. In the multivariate analysis, the presence of intraretinal fluid at year 1 was predictive for VA at 4 years but not before or thereafter. CRT and time to dryness did not predict VA at any of the time points (Table 3). The covariate with the strongest and consistent predictive value for VA at years 2 to 5 was MA at year 1: VA was worse in eyes with MA than in those without MA. The eyes of patients with

**Table 2. Univariate regression analysis of prognostic parameters possibly predicting long term visual acuity.** Parameters with significant predictive power over at least 2 time points were included in the multivariate model.

| | | VA year 2 | | VA year 3 | | VA year 4 | | VA year 5 | |
|---|---|---|---|---|---|---|---|---|---|
| | | p-value for β | $R^2$ | p-value for β | $R^2$ | p-value for β | $R^2$ | p-value for β | $R^2$ |
| Control | VA at BL | **2e-16** | **0.21** | **4.3e-12** | **0.14** | **6.5e-7** | **0.10** | **1.9e-6** | **0.12** |
| | VA change BL–after loading | **0.04** | **<0.01** | 0.27 | <0.01 | 0.56 | <0.01 | 0.51 | <0.01 |
| | VA change BL– 1 year | **8.8e-11** | **0.10** | **1.6e-7** | **0.08** | **7.7e-5** | **0.06** | **5.6e-4** | **0.06** |
| Independent variables | CRT after loading | **0.001** | **0.02** | **0.007** | **0.02** | 0.06 | 0.01 | 0.06 | 0.02 |
| | CRT change BL–after loading | 0.07 | <0.01 | 0.11 | <0.01 | 0.16 | <0.01 | 0.28 | <0.01 |
| | CRT change BL–year 1 | 0.08 | <0.01 | 0.10 | <0.01 | 0.15 | <0.01 | 0.13 | <0.01 |
| | SRF after loading | 0.63 | <0.01 | 0.49 | <0.01 | 0.28 | <0.01 | 0.91 | <0.01 |
| | SRF at year 1 | 0.42 | <0.01 | 0.67 | 0.42 | 0.23 | 0.42 | 0.73 | 0.42 |
| | IRF after loading | **0.03** | **0.01** | 0.09 | <0.01 | **0.01** | **0.02** | **0.02** | **0.03** |
| | IRF at year 1 | **3.3e-4** | **0.03** | **0.002** | **0.03** | **0.002** | **0.04** | **0.03** | **0.02** |
| | IRF and SRF after loading | 0.11 | <0.01 | 0.24 | <0.01 | 0.14 | <0.01 | 0.11 | <0.01 |
| | IRF and SRF at year 1 | 0.12 | <0.01 | 0.07 | <0.01 | 0.20 | <0.01 | **0.036** | **0.02** |
| | Time to dryness | **0.007** | **0.02** | **0.0007** | **0.04** | **0.04** | **0.02** | 0.06 | 0.02 |
| | Interval extension in the 1st year | **0.013** | **0.01** | **0.02** | **0.02** | 0.84 | <0.01 | 0.98 | <0.01 |
| | Interval extension in the 2nd year (< vs. ≥10 weeks) | **7.6e-4** | **0.03** | **5.5e-4** | **0.04** | 0.09 | <0.01 | 0.27 | <0.01 |
| | MA at baseline | 0.46 | <0.01 | 0.17 | 0.02 | 0.74 | <0.01 | 0.88 | <0.01 |
| | MA after loading | 0.12 | 0.02 | **0.03** | **0.08** | 0.28 | <0.01 | 0.75 | <0.01 |
| | MA at year 1 | **4.3e-16** | **0.16** | **1.6e-14** | **0.18** | **5.9e-10** | **0.15** | **5.5e-9** | **0.18** |
| | Nr of injections in the 1st year | 0.38 | <0.01 | 0.39 | <0.01 | 0.45 | <0.01 | 0.93 | <0.01 |

VA: Visual acuity; BL: baseline. VA at the different time points is the dependent variable, whilst the other variables are the independent variables which were tested one by one in a series of univariate regression analyses. $R^2$: proportion of variation explained by the model: proportion of variance in the dependent variable that can be explained by the independent variable. For example, $R^2 = 0.23$ means that 23% of the variation in the dependent variable is explained by the model. CRT: Central retinal thickness; SRF: Subretinal fluid; IRF: intraretinal fluid. MA: Macular atrophy. Marked in **bold** are the statistically significant results (p<0.05).

MA had a mean of 13 letters lower in VA after 2 years, 16.6 letters after 3 years, and 20.1 letters after 5 years.

CRT explained approximately 1 to 2% of the variation in VA, IRF as well as time to dryness explained 2 to 4% of the variation. The interval extension within the 1st or 2nd year moderately predicted between 1 and 4% of variability. All of these parameters represent significant, but small effect sizes. Finally, MA at year 1 after diagnosis reached a clinically relevant effect size, explaining 18% of the long-term visual outcomes. After inclusion of all of the aforementioned parameters, this model explainedwith 23–34% of the variation in VA a relevant effect size.

## Discussion

According to the univariate model, CRT after loading, intraretinal fluid after 1 year and time to dryness, taken alone, predicted, to some extent, explaining between 1 and 8% of the variation in medium- and long-term visual outcomes (from 2 to 5 years). MA alone, in contrast, closely predicted visual outcomes, explaining 15% to 18% of the variation in VA. This effect persisted in the multivariate model for 2 to 5 years. Given the increasing effect of this parameter over time, it seems surprising that MA has not systematically been reported [1, 15, 28] and has not been introduced as an outcome predictor in prospective randomized clinical trials (**Fig**

**Table 3. Results of the multiple linear regression analyses including different predictors for long-term visual acuity (with visual acuity at baseline as control).**

| | t | p | β | F | df | p | adj. $R^2$ |
|---|---|---|---|---|---|---|---|
| **Dependent variable: Visual acuity** | | | | | | | |
| **After 2 years:** | | | | 33.0 | 343 | <2.2e-16** | 0.32 |
| VA at baseline | 7.63 | 2.3e-13*** | 0.47 | | | | |
| CRT after loading | -2.56 | 0.011* | -0.02 | | | | |
| IRF at 1 year | -1.42 | 0.16 | -2.43 | | | | |
| Time to dryness | -0.61 | 0.54 | -0.05 | | | | |
| MA at 1 year | -6.56 | 2.0e-10*** | -13.20 | | | | |
| **After 3 years:** | | | | 22.0 | 269 | <2.2e-16** | 0.28 |
| VA at baseline | 4.60 | 6.4e-6*** | 0.34 | | | | |
| CRT after loading | -1.54 | 0.12 | -0.01 | | | | |
| IRF at 1 year | -1.43 | 0.16 | -2.92 | | | | |
| Time to dryness | -1.67 | 0.10 | -0.15 | | | | |
| MA at 1 year | -6.54 | 3.16e-10*** | -16.69 | | | | |
| **After 4 years:** | | | | 12.5 | 207 | 1.3e-10*** | 0.21 |
| VA at baseline | 2.82 | 0.005** | 0.28 | | | | |
| CRT after loading | -1.20 | 0.23 | -0.01 | | | | |
| IRF at 1 year | -2.33 | 0.021* | -6.25 | | | | |
| Time to dryness | -0.36 | 0.72 | -0.04 | | | | |
| MA at 1 year | -5.16 | 5.8e-7*** | -17.85 | | | | |
| **After 5 years:** | | | | 10.5 | 150 | 1.1e-8*** | 0.24 |
| VA at baseline | 3.10 | 0.0023** | 0.37 | | | | |
| CRT after loading | -0.38 | 0.70 | -0.01 | | | | |
| IRF at 1 year | -1.30 | 0.20 | -4.56 | | | | |
| Time to dryness | -0.45 | 0.65 | -0.06 | | | | |
| MA at 1 year | -4.81 | 3.7e-6*** | -20.76 | | | | |

VA: visual acuity; CRT: central retinal thickness; IRF: intraretinal fluid; MA: macular atrophy

* $p<0.05$

** $p<0.01$

*** $p<0.001$.

**Parameters of the single variables**:

t: t statistic (the coefficient divided by the standard error). It can be thought of as a measure of precision with which the regression coefficient is measured.

p: p value

β: The beta coefficient is the degree of change in the outcome variable for every 1 unit of change in the predictor variable. Each β coefficient represents the change in the mean response, E(y), per unit increase in the associated predictor variable when all the other predictors are held constant.

**Model parameters**:

F: the value of the *F*-statistic (indicates that we are comparing to an F-distribution).

df: degrees of freedom. The regression degrees of freedom are indicated. The second value in parentheses indicates the residual degrees of freedom.

R: correlation coefficient (measure of the strength of association between the 2 variables)

$R^2$: proportion of variation explained by the model: proportion of variance in the dependent variable that can be explained by the independent variable

Adjusted $R^2$: percentage of variation explained by the model in the population.

3). It might be explained by the fact that MA regularly escapes detection at baseline because of overlay phenomena. Introducing MA at year 1 as a covariate in the regression model in our series demonstrated its strong impact on long-term VA development. AMD is known to progress to MA over time, and current evidence indicates that anti-VEGF treatment may partially contribute to its development [29]. In our study, we included MA that was present at the end

of year 1, and did not take into account newly developing MA beyond year 1, which supports the robustness of this finding. Poor visual acuity at baseline, retinal angiomatous proliferation, presence of intraretinal fluid and absence of subretinal fluid, have been linked to MA in a recent prospective study [30].

This model may namely gain interest for the comparison of anti-VEGF drug effects after correcting for baseline functional characteristics, i.e. baseline VA. After correction for baseline VA, we were able to assess the impact of morphological OCT-derived structural parameters, such as the presence of IRF and SRF after the 1st year, as well as MA at the end of year 1, onto the long-term functional outcomes of treatment. If all of the above-mentioned parameters were recorded in a standardized fashion, it would be well conceivable to compare drug effects between studies independent of the treatment demand, including morphological treatment response and drying potential.

A strength of our study is its robust sample size, long follow-up period of 5 and more years, a treatment-agnostic approach, use of robust and easily identified discriminators, and correction for baseline VA. We aimed to exclude a selection bias induced by the exclusion of data from patients not followed for a longer time or due to treatment cessation because of an unsatisfactory functional response by comparing the predictive power for the whole cohort with that of only patients still under treatment, which revealed no difference. On the other hand, our results showed that IRF, but not SRF, after the loading phase and after 1 year is predictive of the visual outcome. However, the robustness of these qualitative findings should be independently confirmed, ideally through semiquantitative analyses based on central retinal fluid volume. The retrospective design, on the other hand, may have resulted in a reduced continuity of the data because not all patients came to their assigned study time points. Therefore, a tolerance of 20% for the choice of clinical examination time points had to be accepted to escape a major number of missing entries or the need to implement the last observation to be carried forward.

## Conclusions

In conclusion, the presence of intraretinal, but not subretinal fluid at the end of the 1st year may predict functional treatment response. MA after the first year of treatment, but not–as might be expected—at baseline or after the loading phase was the strongest independent predictive biomarker for short- and long-term treatment response. This model does not aim to impact treatment strategies but offers a prognostic tool for predicting the long-term functional outcomes in the clinical setting and eventually also in and between studies. Hence, MA alone explains 18%, and after inclusion of further biomarkers in a treatment-agnostic multivariate regression model up to 32% of the variation in visual outcomes.

## Supporting information

**S1 Table. Effect sizes in univariate analyses.**
(DOCX)

**S2 Table. Confidence intervals for the covariates included in the multiple regression model.**
(DOCX)

**S1 Fig. Treatment interval distribution.**
(TIF)

## Author Contributions

**Conceptualization:** Isabel B. Pfister, Christin Schild, Justus G. Garweg.

**Data curation:** Isabel B. Pfister, Christin Schild.

**Formal analysis:** Isabel B. Pfister.

**Funding acquisition:** Justus G. Garweg.

**Methodology:** Isabel B. Pfister, Justus G. Garweg.

**Project administration:** Justus G. Garweg.

**Resources:** Justus G. Garweg.

**Supervision:** Justus G. Garweg.

**Validation:** Isabel B. Pfister.

**Visualization:** Isabel B. Pfister, Christin Schild.

**Writing – original draft:** Isabel B. Pfister, Justus G. Garweg.

**Writing – review & editing:** Isabel B. Pfister, Christin Schild, Justus G. Garweg.

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
