## [Decision Letter · Decision Letter 0]

8 Aug 2024

PONE-D-24-23017Predicting long-term functional anti-VEGF treatment outcomes in a real-world settingPLOS ONE

Dear Dr. Garweg,

Thank you for submitting your manuscript to PLOS ONE. After careful consideration, we feel that it has merit but does not fully meet PLOS ONE’s publication criteria as it currently stands. Therefore, we invite you to submit a revised version of the manuscript that addresses the points raised during the review process.

We look forward to receiving your revised manuscript.

Kind regards,

Tatsuya Inoue

Academic Editor

PLOS ONE

Journal Requirements:

2. Thank you for stating the following in your Competing Interests section:"None of the authors received direct or indirect support for this investigator-initiated study or have any conflicting interests with the data that are presented in this report." 

4. We notice that your supplementary tables are included in the manuscript file. Please remove them and upload them with the file type 'Supporting Information'. Please ensure that each Supporting Information file has a legend listed in the manuscript after the references list.

Reviewers' comments:

Reviewer's Responses to Questions

**Comments to the Author**

1. Is the manuscript technically sound, and do the data support the conclusions?

Reviewer #1: Yes

Reviewer #2: No

2. Has the statistical analysis been performed appropriately and rigorously? 

Reviewer #1: Yes

Reviewer #2: No

3. Have the authors made all data underlying the findings in their manuscript fully available?

Reviewer #1: Yes

Reviewer #2: No

4. Is the manuscript presented in an intelligible fashion and written in standard English?

Reviewer #1: Yes

Reviewer #2: Yes

5. Review Comments to the Author

Reviewer #1: ・The authors described that MA after one year was the strongest predictor. But what caused MA at 1 year? They should rather evaluate the factors which cause MA at 1 year. Also, discuss it in the discussion section.

・Are there patients who received cataract surgery during the follow up? This could influence on the result.

・In conclusion, they described that the number not reaching treatment intervals of >10 weeks may be a robust biomarker for early treatment response and drying potential of anti-VEGF. However, previous studies showed that drying in the induction phase influences on final visual acuity. I feel this result is a little bit controversial. You should discuss it more.

Reviewer #2: The authors have carefully analyzed long-term visual prognostic factors using a large retrospective cohort, revealing that macular atrophy at one year post-treatment initiation holds significant explanatory power for long-term visual prognosis. While many clinical trials on nAMD focus on factors related to exudative control, focusing on atrophy is clinically important, and the results presented using real-world data are both useful and intriguing for clinicians involved in nAMD treatment.

The manuscript is presented in an intelligible fashion. However, authors need to address issues presented below:

• The criteria used by the authors to select covariates for the multivariate linear regression model are ambiguous. The authors have stated in the Methods section, "In cases with several options for the same variable (CRT, IRF and SRF as well as MA), the timepoint with the most explanatory potential (R2) was chosen." However, it remains unclear how the authors would proceed if the relationship of R2 values varied across different dependent variables (VA at each timepoint).

• The authors state, "To exclude a selection bias, we furthermore assessed the subsample of eyes that had been followed over minimally 5 years after diagnosis under intravitreal anti-VEGF therapy (n = 262), undertaking the same analyses, which revealed almost identical results (data not shown)." However, these results should be presented in the Supplementary material.

• In Results, the authors say, “Interval extension to more than 10 weeks was predictive of short-term, but not long-term VA (up to 3 years). We therefore decided not to include this covariate in our model.” However, CRT after loading was also not a significant predictor for VA up to year 3. Even if CRT after loading is a well-known factor, why is there a need to exclude “Interval extension to more than 10 weeks” from the multivariate model just for that reason?

• In Conclusions, “The number of eyes not reaching treatment intervals of ≥10 weeks may be a more robust biomarker for early treatment response and drying potential of anti-VEGF drugs.” This statement seems inconsistent with the previous content. A discussion on why it can be called a robust biomarker is necessary.

• The main finding presented by the authors (MA at 1 year predicting long-term visual acuity) should be discussed in relation to the natural history of AMD or the impacts of treatment. In the Discussion, the authors state, “This might be explained by the fact that MA regularly escapes detection at baseline because of overlay phenomena.” Do the authors believe that MA exists from the baseline? This should be considered since MA after the loading phase showed less predictive power for long-term visual acuity (Table 2). The state of RPE and the outer retina is generally clear after the loading phase. It is a well-known fact that MA appears after nAMD treatment and its causal relationship with anti-VEGF therapy is suspected（Macular Atrophy in Neovascular Age-Related Macular Degeneration: A Randomized Clinical Trial Comparing Ranibizumab and Aflibercept (RIVAL Study) - PubMed (nih.gov)）. How did eyes with MA compare to those without in terms of the number of injections and control of exudation?

• It is mentioned that patients were treated with T & E, but more specific protocols should be provided. How often were the dosing intervals extended? Also, how was the reduction in intervals handled when exudation recurred?

• To ensure that readers fully understand the advantages of this work on real-world data, more detailed descriptions of patient outcomes are necessary. It is mentioned in the results that 39.5% of patients were lost to follow-up. Were the remaining 60.5% continuously treated with T&E for five years? Were there any cases where the treatment method was switched to pro re nata? Were cases with atypical outcomes like macular hemorrhage or RPE tear during follow-up included in the analysis? Were there any cases that received other treatments like PDT?

• The breakdown of nAMD subtypes included in the study should be also mentioned. Eyes that exhibited macular atrophy at one year might be those with a higher risk of atrophy, such as RAP.

• In Results, the authors say, "this model reached an unprecedented effect size," which should be kept to objective expression. Whether the effect size is "unprecedented" can only be understood by the reader if other reports on long-term visual prognosis prediction models are specifically mentioned, demonstrating the model's higher explanatory power.

• In Results, “Covariates with a significant predictive value over at least 2 timepoints were integrated in the model.” For better readability, it should be stated as: “Covariates with a significant predictive value over at least 2 timepoints were integrated into a multivariate model in the second step.”

6. PLOS authors have the option to publish the peer review history of their article (what does this mean?). If published, this will include your full peer review and any attached files.

Reviewer #1: No

Reviewer #2: No

---

## [Author Response · Author response to Decision Letter 0]

17 Sep 2024

Response to the Editor and the Reviewers:

Editor:

Response: 

Confirmed

2. Thank you for stating the following in your Competing Interests section: "None of the authors received direct or indirect support for this investigator-initiated study or have any conflicting interests with the data that are presented in this report." 

Response: 

Done

Response: 

Since this is not a central part of the research, we eliminated this part.

4. We notice that your supplementary tables are included in the manuscript file. Please remove them and upload them with the file type 'Supporting Information'. Please ensure that each Supporting Information file has a legend listed in the manuscript after the references list.

Response: 

done

Reviewer Comments to the Author

1. Is the manuscript technically sound, and do the data support the conclusions?

Reviewer #1: Yes

Reviewer #2: No

Response: 

Thanks, improved as requested by reviewer 2 (see below)

2. Has the statistical analysis been performed appropriately and rigorously? 

Reviewer #1: Yes

Reviewer #2: No

Response: 

Revised as requested

3. Have the authors made all data underlying the findings in their manuscript fully available?

Reviewer #1: Yes

Reviewer #2: No

Response: 

We have added to the method section a link where to find all data and the R syntax

used for the calculations in this manuscript.

Review Comments to the Author

Reviewer #1: ・The authors described that MA after one year was the strongest predictor. But what caused MA at 1 year? They should rather evaluate the factors which cause MA at 1 year. Also, discuss it in the discussion section.

Response: 

We fully agree that in an ideal world influencing and preventing the development of MA would be an important aim of treatment. 

It has remained to be established what drives the fibrotic switch of MMV and to which part the underlying atrophic component and the fibrovascular complex contribute to a change of the cellular micro-environment resulting in photoreceptor and RPE loss, the hallmark of MA, over time in dry, untreated and treated neovascular AMD (Sivaprasad S, Chandra S, Sadda S, Teo KYC, Thottarath S, de Cock E, Empeslidis T, Esmaeelpour M. Predict and Protect: Evaluating the Double-Layer Sign in Age-Related Macular Degeneration. Ophthalmol Ther. 2024 Aug 16. doi: 10.1007/s40123-024-01012-y; Spaide RF. Pathways to Geographic Atrophy in Non-Neovascular Age-Related Macular Degeneration. Retina. 2024 Aug 9. doi: 10.1097/IAE.0000000000004242). Partially, this could be explained by the fact that MA regularly escapes detection at baseline because of overlay phenomena (Cicinelli MV, Barlocci E, Rissotto F, Russo A, Giuffrè C, Introini U, Bandello F. The Discrepancy Between Visual Acuity Decline and Foveal Involvement in Geographic Atrophy. Ophthalmol Retina. 2024 Aug 7:S2468-6530(24)00361-0. doi: 10.1016/j.oret.2024.07.025). Since robust biomarkers for its development and progression are still to be identified, we added a paragraph about evidence-based aspects or risk factors for MA in treated wet AMD in the discussion section (lines 268-271): “Given the increasing effect of this parameter over time, it seems surprising that MA has rarely been reported1,15,28 and has not been introduced as an outcome predictor in prospective randomized clinical trials. This might be explained by the fact that MA regularly escapes detection at baseline because of overlay phenomena. Introducing MA at year 1 as a covariate in the regression model in our series demonstrated the degree of its impact on long-term VA development. Independent of treatment, the natural evolution of AMD leads to a progression to MA, but evidence indicates that anti-VEGF treatment contributes to its development and progression (Foss, Rotsos, Empeslidis & Chong, Development of macular atrophy in patients with wet age-related macular degeneration receiving anti-VEGF treatment. Ophthalmologica, 2022;245:204–217, DOI: 10.1159/000520171). Here, we included MA that was present at the end of year 1, and did not take into account newly developing MA after year 1, which supports the robustness of this finding. Possible risk factors could be poor visual acuity at baseline, retinal angiomatous proliferation, presence of intraretinal fluid and absence of subretinal fluid, as reported in a recent prospective study (Gemenetzi, M., A. J. Lotery, and P. J. Patel. "Risk of geographic atrophy in age-related macular degeneration patients treated with intravitreal anti-VEGF agents." Eye 31.1 (2017): 1-9). 

We have therefore speculated that MA may have been present at baseline but escaped detection in some cases. Therefore, we went back to the OCT images of all patients with MA present at year 1 to reassess the presence of MA at baseline and after the loading phase and found that 43 eyes out of 433 (10%) showed some MA at baseline and 14% after the loading phase. Linear regression analyses (Table 2) shows, that MA at baseline or after the loading did not predict VA. 

Are there patients who received cataract surgery during the follow up? This could influence on the result.

Response: 

This is indeed an interesting point that was missing and has been added now. A total of 108 out of 433 eyes (24.9%) had a cataract surgery after baseline (start with anti-VEGF treatment). We compared these eyes to those which were pseudophakic at baseline or remained phakic during the study period regarding MA after 1 year and visual acuity over the observation period. There was no significant difference for any of these, and we found no correlation between cataract surgery during follow up and atrophy after 1 year (r=-0.052, p=0.30). We correspondingly added a section in results (lines 138-140).

 MA yes No MA Missing

No cat surgery during follow up 51 (15.7%) 250 (76.9%) 24 (7.4%)

With cat surgery during follow up 13 (12%) 90 (83.3%) 5 (4.6%)

Chi-square test p=0.30

 No cat surgery during follow up

(mean±SD) With cat surgery during follow up (mean±SD) Mann-Whitney U test

ETDRS at baseline 64.8±12.8 62.8±13.8 p=0.25

ETDRS after loading 70.3±11.4 67.5±13.0 p=0.06

ETDRS after 1 year 70.7±13.8 68.2±12.6 p=0.01

ETDRS after 2 years 68.3±17.5 68.2±14.5 p=0.22

ETDRS after 3 years 66.8±18.4 68.2±15.1 p=0.96

ETDRS after 4 years 62.2±21.8 66.6±15.9 p=0.40

ETDRS after 5 years 62.7±23.1 65.5±18.0 p=0.94

In conclusion, they described that the number not reaching treatment intervals of >10 weeks may be a robust biomarker for early treatment response and drying potential of anti-VEGF. However, previous studies showed that drying in the induction phase influences on final visual acuity. I feel this result is a little bit controversial. You should discuss it more.

Response: 

We agree that time to dryness and outcomes of treatment interval extension were not analysed in this study. Therefore, we decided to delete this personal opinion because of lack of sufficient evidence. 

Reviewer #2: The authors have carefully analyzed long-term visual prognostic factors using a large retrospective cohort, revealing that macular atrophy at one year post-treatment initiation holds significant explanatory power for long-term visual prognosis. While many clinical trials on nAMD focus on factors related to exudative control, focusing on atrophy is clinically important, and the results presented using real-world data are both useful and intriguing for clinicians involved in nAMD treatment. 

The manuscript is presented in an intelligible fashion. However, authors need to address issues presented below:

• The criteria used by the authors to select covariates for the multivariate linear regression model are ambiguous. The authors have stated in the Methods section, "In cases with several options for the same variable (CRT, IRF and SRF as well as MA), the timepoint with the most explanatory potential (R2) was chosen." However, it remains unclear how the authors would proceed if the relationship of R2 values varied across different dependent variables (VA at each timepoint).

Response: 

Out of these parameters, for our multivariate analyses we chose the ones which maximally explained the variance in all cases. This has been added to the methods sections (lines 126-28).

This is displayed in table 2: the results of the univariate regression analyses reveal, which of the options had to be selected. We never observed that the relationship of R2 values varied across different dependent variables:

Out of 3 options for VA, for example, 2 were significant at all 4 timepoints. VA at BL explains over 10% of variation in all 4 time points, whereas the other option (VA change from BL to 1 year) explains 10% of variation after the first year, but less than 10% at the other time points.

For CRT only one of the 3 options showed a significant result at 2 of the four time points. Here we admit that it was not obvious to take CRT into the model instead of interval extension beyond 10 weeks. We therefore added this to the model.

For SRF, none of the 2 options revealed a significant signal at any point of the time. For IRF we had 2 possible options: IRF after loading or IRF after 1 year. IRF after loading showed a significant result at 3 out of 4 timepoints whereas IRF after 1 year significantly explained variation in VA at all 4 time points. For MA, only one of the 3 options clearly explained the variation in VA at all 4 time points, leaving no doubt to which option to choose (MA at 1 year).

• The authors state, "To exclude a selection bias, we furthermore assessed the subsample of eyes that had been followed over minimally 5 years after diagnosis under intravitreal anti-VEGF therapy (n = 262), undertaking the same analyses, which revealed almost identical results (data not shown)." However, these results should be presented in the Supplementary material.

Response: 

As stated above, since this is not a central part of our results, we eliminated this part. We can of course add it, if the reviewers think this would be an important information adding to the validity of the study.

• In Results, the authors say, “Interval extension to more than 10 weeks was predictive of short-term, but not long-term VA (up to 3 years). We therefore decided not to include this covariate in our model.” However, CRT after loading was also not a significant predictor for VA up to year 3. Even if CRT after loading is a well-known factor, why is there a need to exclude “Interval extension to more than 10 weeks” from the multivariate model just for that reason?

Response: 

Thank you for pointing on this issue. We now extended our model to include the variable “Interval extension to 10 or more weeks in the 2. year”. As shown in Table 2, this variable shows a similar explanatory potential as CRT after the loading.

• In Conclusions, “The number of eyes not reaching treatment intervals of ≥10 weeks may be a more robust biomarker for early treatment response and drying potential of anti-VEGF drugs.” This statement seems inconsistent with the previous content. A discussion on why it can be called a robust biomarker is necessary.

Response: 

We fully agree with the reviewer that this was a personal opinion which was not supported by the results mentioned above. Therefore, this sentence was removed.

• The main finding presented by the authors (MA at 1 year predicting long-term visual acuity) should be discussed in relation to the natural history of AMD or the impacts of treatment. In the Discussion, the authors state, “This might be explained by the fact that MA regularly escapes detection at baseline because of overlay phenomena.” Do the authors believe that MA exists from the baseline? This should be considered since MA after the loading phase showed less predictive power for long-term visual acuity (Table 2). The state of RPE and the outer retina is generally clear after the loading phase. It is a well-known fact that MA appears after nAMD treatment and its causal relationship with anti-VEGF therapy is suspected（Macular Atrophy in Neovascular Age-Related Macular Degeneration: A Randomized Clinical Trial Comparing Ranibizumab and Aflibercept (RIVAL Study) - PubMed (nih.gov). How did eyes with MA compare to those without in terms of the number of injections and control of exudation?

Response: 

We thank the reviewer for having requested to look at the possible role of anti-VEGF therapy and the mechanisms underlying MA at year one. 

It has remained to be established to which part the underlying atrophic component and the fibrovascular complex contribute to a change of the cellular micro-environment resulting in photoreceptor and RPE loss, the hallmark of MA, over time in dry, untreated and treated neovascular AMD (Sivaprasad S, Chandra S, Sadda S, Teo KYC, Thottarath S, de Cock E, Empeslidis T, Esmaeelpour M. Predict and Protect: Evaluating the Double-Layer Sign in Age-Related Macular Degeneration. Ophthalmol Ther. 2024 Aug 16. doi: 10.1007/s40123-024-01012-y; Spaide RF. Pathways to Geographic Atrophy in Non-Neovascular Age-Related Macular Degeneration. Retina. 2024 Aug 9. doi: 10.1097/IAE.0000000000004242). Partially, this could be explained by the fact that MA regularly escapes detection at baseline because of overlay phenomena (Cicinelli MV, Barlocci E, Rissotto F, Russo A, Giuffrè C, Introini U, Bandello F. The Discrepancy Between Visual Acuity Decline and Foveal Involvement in Geographic Atrophy. Ophthalmol Retina. 2024 Aug 7:S2468-6530(24)00361-0. doi: 10.1016/j.oret.2024.07.025). Since robust biomarkers for its development and progression are still to be identified, we added a paragraph about evidence-based aspects or risk factors for MA in treated wet AMD in the discussion section (lines 268 ff): “Given the increasing effect of this parameter over t

---

## [Decision Letter · Decision Letter 1]

6 Nov 2024

Predicting long-term functional anti-VEGF treatment outcomes in neovascular AMD in a real-world setting

PONE-D-24-23017R1

Dear Dr. Garweg,

We’re pleased to inform you that your manuscript has been judged scientifically suitable for publication and will be formally accepted for publication once it meets all outstanding technical requirements.

Kind regards,

Tatsuya Inoue

Academic Editor

PLOS ONE

Additional Editor Comments (optional):

The authors addressed all the comments.

Reviewers' comments:

Reviewer's Responses to Questions

**Comments to the Author**

1. If the authors have adequately addressed your comments raised in a previous round of review and you feel that this manuscript is now acceptable for publication, you may indicate that here to bypass the “Comments to the Author” section, enter your conflict of interest statement in the “Confidential to Editor” section, and submit your "Accept" recommendation.

Reviewer #2: All comments have been addressed

2. Is the manuscript technically sound, and do the data support the conclusions?

Reviewer #2: Yes

3. Has the statistical analysis been performed appropriately and rigorously? 

Reviewer #2: Yes

4. Have the authors made all data underlying the findings in their manuscript fully available?

Reviewer #2: Yes

5. Is the manuscript presented in an intelligible fashion and written in standard English?

Reviewer #2: Yes

6. Review Comments to the Author

Reviewer #2: (No Response)

7. PLOS authors have the option to publish the peer review history of their article (what does this mean?). If published, this will include your full peer review and any attached files.

Reviewer #2: **Yes: **Shuichiro Aoki

---

## [Editor Report · Acceptance letter]

14 Nov 2024

PONE-D-24-23017R1 

PLOS ONE

Dear Dr. Garweg, 

I'm pleased to inform you that your manuscript has been deemed suitable for publication in PLOS ONE. Congratulations! Your manuscript is now being handed over to our production team.

Kind regards, 

on behalf of

Dr. Tatsuya Inoue 

Academic Editor

PLOS ONE